# Some Biomechanical and Anthropmetric Differences Between Elite Swimmers with Down Syndrome and Intellectual Disabilities

**DOI:** 10.3390/sports14010028

**Published:** 2026-01-06

**Authors:** Ana Querido, António R. Sampaio, Ana Silva, Rui Corredeira, Daniel Daly, Ricardo J. Fernandes

**Affiliations:** 1Research Center of the Polytechnic Institute of Maia (N2i), Maia Polytechnic Institute (IPMAIA), Castêlo da Maia, 4475-690 Maia, Portugal; arsampaio@ipmaia.pt; 2Centre of Research, Education, Innovation and Intervention in Sport (CIFI2D), Faculty of Sport, University of Porto, 4200-450 Porto, Portugal; ricfer@fade.up.pt; 3Porto Biomechanics Laboratory (LABIOMEP-UP), Faculty of Sport, University of Porto, 4200-450 Porto, Portugal; 4Research Center in Sports Sciences, Health Sciences and Human Development (CIDESD), University of Maia (UMaia), 4475-690 Maia, Portugal; 5Sport Physical Activity and Health Research & Innovation Center (SPRINT), 4900-347 Rio Maior, Portugal; abragasilva@esdl.ipvc.pt; 6Research Centre in Physical Activity, Health and Leisure, CIAFEL, Faculty of Sport, University of Porto, 4200-450 Porto, Portugal; rcorredeira@fade.up.pt; 7Faculty of Movement and Rehabilitation Sciences, Katholieke Universiteit Leuven, 3001 Leuven, Belgium; daniel.daly@kuleuven.be

**Keywords:** kinematics, coordination, anthropometrics, swimming

## Abstract

The purpose was to characterize and compare biomechanical and coordinative parameters at maximum velocity between swimmers with Down syndrome and intellectual disabilities and examine these in relation to their anthropometrics. Nine swimmers (four with Down syndrome and five with intellectual disabilities) performed three bouts of 25 m crawl stroke, each at maximum velocity, which were recorded with the Qualysis motion analysis system. Anthropometric variables, BMI, and percentage of body fat were also assessed. Swimmers with Down syndrome presented a smaller height, acromion height, sitting height, arm span, hand length, hand width, foot length, foot width, and velocity than swimmers with intellectual disabilities. Swimmers with Down syndrome have disadvantageous anthropometrics and slower swimming velocities compared to swimmers with intellectual disabilities. Those swimmers also appear to present distinctive coordination (catch-up for Down syndrome and superposition for intellectual disabilities) and intracyclic velocity variation (Down syndrome presented higher values) compared to swimmers with intellectual disabilities, suggesting a lower swimming efficiency.

## 1. Introduction

Athletes with intellectual disabilities were re-introduced into the Paralympic Games in London 2012 after being excluded at the end of the Sidney 2000 Paralympics [1]. In the meantime, the classification system has been changed in a way that demonstrates the impact of a particular impairment on a specific sport, classifying the athletes according to the impact severity so that competition is based on skill, training, and effort and not on the disability level [2]. Naturally, the question of what evidence can demonstrate the impact of intellectual disability on high-level performance is still somewhat open [1].

Down syndrome is a genetic condition that can be shown by the karyotype, presenting a unique etiology and affecting many areas of development. These may result in changes in particular biomechanical, physiological, anatomical, and behavioral characteristics, with evident repercussions on the health and social contexts of persons with this condition [3,4,5]. The incidence of trisomy is influenced by maternal age and differs in population, with an estimated live birth prevalence of 10.1 per 10,000 live births in Europe [6,7]. Intellectual disability is characterized by significant limitations both in intellectual functioning and adaptative behavior, as expressed in terms of conceptual, social, and practical adaptive skills [8]. Due to the nature of intellectual disabilities, motor function is also often impaired, requiring tailored physical developmental strategies [9].

Aquatic exercise has been shown to offer benefits for people with intellectual disabilities in terms of cardiorespiratory endurance, muscular endurance, speed, static balance, and agility [10,11], and may be an attractive alternative to land-based exercise for individuals with musculoskeletal conditions such as low muscle tone and excess adiposity, as found in adults with Down syndrome [12]. There has been increasing interest from people with Down syndrome in competitive swimming [13]. Swimmers with Down syndrome can compete at the International Paralympic Committee (IPC) events included in the S14 Class, dedicated to swimmers with intellectual disabilities. Interestingly, as no swimmers with Down syndrome have yet achieved a level of performance that allows them to participate in a Paralympic event, it seems unfair that swimmers with Down syndrome compete in the same class as swimmers with intellectual disabilities.

Studies concerning competitive-level persons with intellectual disabilities (and also Down syndrome) are still scarce [12,14], and research on the effects of swimming on swimmers with intellectual disabilities is limited. A notable gap in exploring the specific challenges and adaptations required for this population in competitive swimming is observed [8]. Although people with intellectual disabilities benefit from improved cardiorespiratory and muscular endurance, speed, balance, and agility in the water, they show distinctive differences in swimming performance and muscular strength [15]. Persons with Down syndrome are known to have poorer strength, cardiovascular fitness, and body composition than persons with intellectual disabilities or non-disabled individuals [16,17,18], although they are able to improve their physical and functional fitness [13,19,20,21,22,23,24,25]. Also, elite swimmers with intellectual disabilities face challenges in speed, particularly during turns, compared to swimmers without intellectual disabilities [26]. These swimmers often encounter challenges in adopting a strategic approach during competitive events, which can adversely impact the performance [8].

In swimming, in addition to the swimmer’s fitness profile, it is important to evaluate their technical level, as velocity (V) depends on stroke rate (SR) and stroke length (SL), and there is a necessity to find the optimal compromise between these parameters to attain and maintain their optimal velocity [27]. Also, the temporal organization of the upper limb cycle is important to characterize highly skilled performance [28], with the index of coordination (IdC) appearing as a measure of the lag time between the propulsive actions of the upper limbs [29]. This index is based on the quantification of the phases of the upper limb action, being a useful tool to assess swimmers’ skill levels [30,31]. Although coordination has not been directly related to swimming propulsion, its connection to buoyancy, breathing, and minimizing drag is fundamental to optimizing propulsive forces [32]. Studies have shown that, when increasing their speed, swimmers should favor a continuity between the left and right upper limb propulsive phases, so they can promote temporal continuity in propulsions (e.g., [29,33]). Swimmers should also be able to adapt their coordination to minimize energy cost [34].

Closely related to the swimmers’ ability to coordinate their propulsive forces are the fluctuations of the instantaneous velocity during a total cycle [35], commonly known as intracyclic velocity variation (IVV). In the front crawl technique, a highly organized inter-arm coordination can minimize IVV and increase swim efficiency, since a higher temporal continuity between propulsive actions minimizes the deceleration between two propulsive cycles [36]. Also, maintaining a lower IVV helps to decrease the energy expenditure [27]. In this way, IdC evaluates how upper limb propulsive actions are distributed over time, with IVV being the kinematical consequence of the distribution of propulsion in time [37]. Previous pilot studies of swimmers with Down syndrome showed their poorer coordinative development and, consequently, their lower technical efficiency compared to swimmers without disabilities [38,39], emphasizing the need to perform more studies on swimmers with this condition, comparing them with swimmers with other intellectual disabilities.

The aims of this study were to (i) characterize, from a kinematical approach, the swimming performance biomechanical and coordinative determinants of swimmers with Down syndrome and intellectual disabilities at maximum velocity and (ii) contribute to the understanding and development of high-level competitive swimming for disabled persons, analyzing the possible differences in the swimming biomechanics of persons with Down syndrome and intellectual disabilities. It was hypothesized that (i) swimmers with intellectual disabilities are faster than those with Down syndrome; (ii) swimmers with Down syndrome present higher SR and SL values than their intellectually disabled peers; and (iii) swimmers with Down syndrome apply different coordination modes and present a higher IVV and intra-variability than swimmers with intellectual disabilities for velocity, SR, SL, IdC, and IVV.

## 2. Materials and Methods

### 2.1. Participants

Nine trained swimmers were divided into two groups according to their national classification: swimmers with Down syndrome (S21; *n* = 4; 25.0 ± 6.2 years old) and swimmers with an intellectual disability (S14; *n* = 5; 18.0 ± 2.0 years old). Swimmers from both groups were engaged in competitive swimming for at least three years; all took part in national competitions, and some took part in international competitions for swimmers with intellectual disabilities (Virtus) and Down syndrome (DSISO). All individuals or their parents gave written informed consent to participate in the current study, which was approved by the local ethics committee (under the code: CEFAD 19.2020) and carried out according to the Declaration of Helsinki.

### 2.2. Experimental Procedure

Testing took place in a 25 m indoor pool, 1.90 m deep, with a water temperature of 27.5 °C. After a moderate-intensity individual warm-up (10 min free swimming), participants performed a 3 × 25 m front crawl maximal effort (30 min rest), from a push-off start, without breathing from the 7.5 to the 20 m marks. The calibrated volume was defined using three calibrations—underwater, overwater, and twin (to merge the first and the latter)—according to the manufacturer’s guidelines. This approach enabled the creation of 3D dual-media volume, where the orthogonal axes were defined as x, y, and z for horizontal, mediolateral, and vertical movements (z = 0 defines water surface). A 13-camera setup (Motion Capture, Qualysis Cameras; Qualisys AB, Gothenburg, Sweden) was used, with seven dry-land and six underwater cameras (Oqus 3+ and Oqus Underwater; Qualisys AB, Gothenburg, Sweden) operating at 100 Hz (Figure 1).

Data acquisition was performed with Qualisys Track Manager version 2.7 (Qualisys AB, Gothenburg, Sweden), and data post-processing employed Visual3D (C-Motion, Germantown, MD, USA) using a low-pass digital filter of 6 Hz. Qualisys is a certified company with an ISO 9001 Certificate and EC Certificate–Medical Device Directive (MDD) that can be found at www.qualisys.com, accessed on 20 July 2025. Each swimmer was equipped with a full body retro-reflective marker setup situated at the anatomical reference points selected, particularly the acromion, lateral and medial humerus epicondyle, radius- and ulna-styloid processes, third distal phalanx, iliac crest, and anterior and posterior iliac spine for both right and left sides.

In addition, anthropometric variables were measured with a stadiometer Seca 213 (Seca GmbH & Co., KG, Hamburg, Germany), and body composition was assessed with a Tanita Inner scan, BC-532 (Tanita, Hoofddorp, The Netherlands) with swimmers wearing light clothes without shoes.

All parameters were calculated as the mean of two recorded upper limb cycles, with one cycle considered from the right or left fingertip entry until the same fingertip re-entry. Horizontal swimming velocity was calculated by the mean frame-by-frame displacement of the right hip over one cycle divided by the cycle time, SR was determined by the upper limb cycle duration, and SL was obtained from the horizontal displacement of the hip during an upper limb cycle. IVV was determined by the coefficient of variation, expressed in terms of percentage [28] and represented as the standard deviation of the instantaneous velocity data divided by the mean velocity of the swimmers’ speed during each repetition, multiplied by 100.

IdC was calculated accordingly to Chollet et al. [29], who divided each upper limb cycle into the following phases: (i) entry and catch, corresponding to the time between the entry of the hand into the water and the beginning of its backward movement; (ii) pull, corresponding to the time between the beginning of the backward movement of the hand and its entry into the plane vertical to the shoulder; (iii) push, corresponding to the time between the positioning of the hand below the shoulder and its exit from the water; and (iv) recovery, corresponding to the time between the exit of the hand from the water and its following entry. IdC gives the time gap between the propulsion of the two upper limbs as a percentage of the duration of the complete upper limb cycle and was the mean of IdCleft and IdCright [29]:IdC_left_ = [(Time_end of push for left arm_ − Time_beginning of pull for right arm_) × 100]/Duration_complete cycle_IdC_right_ = [(Time_end of push for right arm_ − Time_beginning of pull for left arm_) × 100]/Duration_complete cycle_

### 2.3. Statistical Analysis

Descriptive statistics (means and standard deviations) were calculated for all the variables, and all data were checked for normality (Shapiro–Wilk test) and homogeneity of variance (Levene’s test). The Kruskal–Wallis test was used to identify the differences between groups in body composition measures, and a Repeated Measures (ANOVA) was conducted to make pairwise comparisons within subjects’ effects and interaction effects. Size effect thresholds were defined as small—0.01, medium—0.06, and large—0.14. The significance level in all analyses was set at 0.05. Statistical analyses were conducted using IBM SPSS software version 21.0 (SPSS, Chicago, IL, USA).

## 3. Results

The swimmers with Down syndrome (S21) were older than the swimmers with an intellectual disability (S14), but were engaged in a similar number of training hours per week. Regarding the anthropometric characteristics, the swimmers with S21 seemed to be at a disadvantage condition compared to S14, since they presented a lower height, acromion and sitting height, arm span, and hand/foot length/width (Table 1). Swimmers with S14 were heavier than S21, and no differences between the groups were noticed for BMI, body fat percentage, or waist–hip ratio.

Differences between S21 and S14 swimmers were observed in swimming speed, with the S14 swimmers being faster (Table 2). As for coordination, despite no differences observed, the S21 swimmers presented a negative IdC, corresponding to a “catch-up” coordination mode, while the coordination of the S14 group was higher than zero, meaning a “superposition” coordination mode. There were also no differences between groups for IVV, and the three repetitions were not different from each other for all variables and both groups. Differences between groups (Down syndrome and intellectual disability) can be observed in velocity at the second and third repetitions, as well as for the IdC at the third repetition.

## 4. Discussion

The main hypotheses of the present study were that (i) swimmers with intellectual disabilities are faster than those with Down syndrome; (ii) swimmers with Down syndrome present higher SR and SL values than their intellectually disabled peers; and (iii) swimmers with Down syndrome apply different coordination modes and present a higher IVV and intra-variability than swimmers with intellectual disabilities for velocity, SR, SL, IdC, and IVV.

To better understand the technical characteristics of swimmers with Down syndrome by comparing them to swimmers with intellectual disabilities, several parameters were evaluated in a maximum velocity test, with three repetitions. Additionally, body measurements were performed, with swimmers with Down syndrome presenting lower anthropometric measures than swimmers with intellectual disabilities, meaning that swimmers with this condition in the present study presented a clear disadvantage in competitive swimming practice. Swimmers with Down syndrome presented a lower height, arm span, and hand/foot length/width. For swimming, these anthropometric characteristics can create hydrodynamic advantages by reducing drag and increasing propulsion [39]. Also, a lower arm span implies a lower distance traveled per stroke [40].

In fact, swimming performance is known to result from a multifactorial process that involves several scientific domains, such as hydrodynamics, kinematics, energetics, and also anthropometrics [41]. The movement patterns of persons with Down syndrome are conditioned, especially due to problems in muscle and tone control, decreased strength and anthropometric traits, and smaller stature and high percentage of body fat [42,43], all conditions that may affect their swimming performance. In fact, the significant influence of some anthropometric parameters is well established, such as hand and foot areas, leg and arm lengths, and height on stroke length, stroke rate, and speed [44]. Swimmers with intellectual disabilities are taller and, as a consequence, present greater body measurements than swimmers with Down syndrome.

A systematic review concerning anthropometrics, stroking parameters, and the performance of young and adolescent swimmers [45] stated that, for the front crawl, arm span and height were the most commonly observed variables related to performance, and the anthropometric characteristics were associated with biomechanical variables. Also, sitting height had a positive effect on front crawl performance. The relationship between sitting height and performance can be explained by the fact that swimmers with a longer torso are also taller, and these characteristics that promote a decrease in the Froud number and in wave-making resistance allow them to cut the water with less resistance and obtain an automatic edge from their longer bodies [46,47].

Also, the advantages of having a greater hand width and hand and foot surface areas and lengths, with a positive relationship between hand surface area and front crawl thrust, have been reported before [45]. So, swimming performance is positively affected by higher propulsive surface areas, which increase hydrodynamic lift force to propel the swimmer through the water, while also allowing them to perform fewer upper limbs cycles for the same distance [48]. A consistent association between arm span, height, upper and lower limb lengths, and stroke length has been observed [45]. Swimmers with the highest height seem to have a higher arm span and surface area as well, such as in the present study, meaning that such swimmers can achieve a higher stroke length and a consequently higher velocity [49,50]. Despite differences in the anthropometric variables and swimming speed between the swimmers with intellectual disabilities and Down syndrome, no differences were observed in the stroke length of both groups. Nevertheless, swimmers with intellectual disabilities presented higher values for the stroke length, and the fact that those differences were not considered significant may be related to the small sample size. Also, the swimmers with Down syndrome trained 2.5 h/week more than the swimmers with intellectual disabilities, so they can be at a slightly higher level of training.

Although there has been an increasing interest from people with Down syndrome in competitive swimming [13], it being a physical activity and sport meaningful for so many people, including those with intellectual disabilities [51], there is still a great lack of research focused on trained individuals [14]. More recently, a study checked for adaptations to swimming training in athletes with Down syndrome [12] after an intervention of 18-week periodized swimming training. These swimmers improved their swimming performance, with less clear changes in body composition and jump performance. All the swimmers participated in adapted swimming competitions at a regional and national level. Nevertheless, despite being considered trained sub-elite swimmers, they trained considerably less hours a week than the swimmers from the present study, and they were considered overweight at baseline, with no significant changes at the end of the intervention program. Unlike the swimmers from the referred above study, the swimmers with Down syndrome from the present study presented BMI values at the normal weight range and body fat percentages similar to swimmers with intellectual disabilities. We may therefore conclude that these swimmers were more highly trained than those from the González-Ravé et al. [12] study.

For the body composition measurements, there were no differences in BMI, percentage of body fat, and waist–hip ratio between swimmers with Down syndrome and intellectually disabled ones in the current study, which is a good indicator for swimmers with Down syndrome, as persons with this condition are often referred to as presenting high levels of BMI and fat percentage [51]. In fact, the two groups from the current study present acceptable BMI values according to the World Health Organization [52], being at the normal weight range, defined between 18.5 and 24.9%.

The swimmers with Down syndrome were found to have coordination patterns similar to those obtained by lower-proficiency swimmers or younger ones, evidenced by their catch-up coordination mode [38,39]. A negative index of coordination was also found in the current study for swimmers with Down syndrome, whilst, for swimmers with intellectual disabilities, the mean IdC was on the superposition mode (IdC > 0%), although there were no differences between the two groups for the mean IdC. Nevertheless, there was a difference at the third repetition of 25 m, which was the only observed difference between the groups, aside from swimming velocity. The literature states that swimmers without disabilities usually change their coordination mode from catch-up to opposition or superposition with an increased velocity [40,53]. In accordance with this, swimmers with intellectual disabilities from the present study presented positive IdC values, meaning that they have chances to have a more proficient technique than swimmers with Down syndrome. Nevertheless, Silva et al. [34] highlighted the fact that upper limb coordination varies with organismic, task, and environmental constraints and suggested that there is no ideal or optimal coordination pattern that youth, learners, and less skilled swimmers should imitate. They suggest that several motor solutions should be adopted based on how each swimmer deals with specific constraints. Obviously, this raises the question of whether swimmers with intellectual disabilities are able to identify and recognize these constraints.

Typically, the fastest swimmers present higher IdC values for equivalent paces and, on the other hand, the IdC increases with fatigue as velocity decreases [27,29], meaning that high IdC values should not be automatically linked with high velocities but instead should be associated with propulsive efficiency indicators such as SL or IVV [28,40]. IVV and IdC provide relevant complementary information to differentiate swimmers according to their skill level [37,40]. An increased IdC with swimming velocity has been suggested to be a strategy adopted by high-level swimmers to maintain a constant IVV [24]. The IVV percentages found in the current study are not different for the two groups, but swimmers with Down syndrome seem to present a somewhat higher velocity variation. Nevertheless, the interaction between parameters such as SL, IdC, and IVV should be better for swimmers with intellectual disabilities, as they swim faster than swimmers with Down syndrome.

Interestingly, there were no differences found in either group for the three repetitions, meaning that the swimmers with Down syndrome and swimmers with intellectual disabilities did not differ in intra-variability for the different analyzed variables. As persons with Down syndrome have combined cognitive and physical limitations, this may contribute to a higher motor performance dysfunction [54,55]; a higher variability would be expected. On the other hand, these swimmers are, to the best of our knowledge, the most highly trained participants with Down syndrome in a study of this nature, meaning that this can be a contributing factor for the minor intra-variability observed.

For further research in the field of swimming for intellectual disabilities and Down syndrome, it is fundamental that the sample sizes should be augmented, as it is a conditioning factor that may affect the final results. This is a limitation from the present study that the authors acknowledge. It would be important to analyze gender differences. Nevertheless, it is difficult to find a sufficient number of swimmers with Down syndrome and intellectual disabilities that train at a high level to this purpose. Despite the fact that there were no differences found between swimmers with Down syndrome and swimmers with intellectual disabilities for most of the variables, the lower velocities found in swimmers with Down syndrome could be at least partially explained by their poorer anthropometric characteristics as well as the less efficient relations between coordinative and velocity variations aspects, which reduce their propulsive times.

## 5. Conclusions

The main conclusions were that (i) swimmers with Down syndrome have disadvantaged anthropometric characteristics compared to swimmers with intellectual disabilities, although no differences in BMI and fat percentage were found; (ii) swimmers with Down syndrome swim at a considerably slower velocity than swimmers with intellectual disabilities; (iii) there were no significant differences between swimmers with Down syndrome and swimmers with intellectual disabilities in SR and SL; (iv) despite no differences in IdC and IVV, swimmers with Down syndrome presented a different coordination mode than swimmers with intellectual disabilities (catch-up vs. superposition) and a higher IVV; the higher intra-variability of swimmers with Down syndrome was not confirmed; and (v) the small sample size might be covering up some significant results.

It is important that coaches acknowledge the importance of these biomechanic and coordinative parameters and use them when preparing their training sessions. Swimmers with Down syndrome and intellectual disabilities should have their SR and SL relationship optimized by training, and it seems important that, especially for swimmers with Down syndrome, coordination and IVV should not be neglected by coaches, as they may make a difference regarding swimming velocity.

## Figures and Tables

**Figure 1 sports-14-00028-f001:**
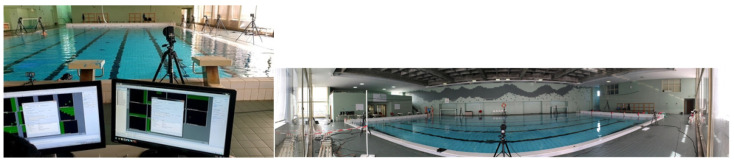
Experimental setup at the swimming pool.

**Table 1 sports-14-00028-t001:** Mean ± SD values for the anthropometric and training background characteristics of swimmers with Down syndrome (S21) and intellectual disabilities (S14).

Parameters	S21 (*n* = 4)	S14 (*n* = 5)
Height (cm)	147.6 ± 5.7 *	174.0 ± 7.9
Acromion height (cm)	120.1 ± 6.2 *	141.8 ± 5.0
Sitting height (cm)	78.2 ± 2.5 *	90.6 ± 4.7
Arm span (cm)	138.8 ± 6.2 *	174.6 ± 8.7
Hand length (cm)	15.2 ± 0.7 *	17.9 ± 0.6
Hand width (cm)	7.5 ± 0.3 *	8.4 ± 0.3
Foot length (cm)	19.4 ± 1.0 *	23.5 ± 1.3
Foot width (cm)	7.7 ± 0.5 *	9.2 ± 0.2
Body mass (kg)	51.7 ± 9.3 *	66.3 ± 3.1
BMI (kg/m2)	23.1 ± 3.4	22.1 ± 2.9
PBF (%)	24.1 ± 12.7	20.7 ± 10.1
WHR	0.9 ± 0.0	0.8 ± 0.0
Training sessions a week (h)	12.5 ± 1.9	10.0 ± 1.4

BMI = body mass index, PBF = body fat percentage, WHR = waist–hip ratio. Differences between groups are identified by *.

**Table 2 sports-14-00028-t002:** Mean ± SD for the stroke and coordinative characteristics of swimmers with Down syndrome (S21) and intellectual disabilities (S14). *p* values for the group differences, repetition differences for each group, and interactions between groups and repetitions.

	S21 (*n* = 4) Mean ± SD				S14 (*n* = 5) Mean ± SD				Group S21 vs. S14			Repetition			Group * Repetition
	1	2	3	T	1	2	3	T	1	2	3	1 vs. 2	1 vs. 3	2 vs. 3	
Speed (m/s)	1.04 ± 0.21	1.06 ± 0.15	1.05 ± 0.11	1.05 ± 1.13 *	1.32 ± 0.23	1.31 ± 0.13	1.35 ± 0.22	1.33 ± 1.19	0.098	0.02 *	0.04 *	S21 = 1.000 S14 = 1.000	S21 = 1.000 S14 = 1.000	S21 = 1.000 S14 = 1.000	0.898
SR (cycles/min)	49.65 ± 11.09	45.09 ± 7.81	42.99 ± 9.60	45.91 ± 8.77	50.07 ± 8.76	53.22 ± 4.61	52.24 ± 3.91	51.84 ± 5.52	0.951	0.091	0.088	S21 = 0.194 S14 = 0.403	S21 = 0.369 S14 = 1.000	S21 = 1.000 S14 = 1.000	0.071
SL (m/cycle)	1.31 ± 0.27	1.33 ± 0.17	1.33 ± 0.24	1.32 ± 0.22	1.51 ± 0.08	1.49 ± 0.01	1.52 ± 0.09	1.51 ± 0.05	0.144	0.072	0.134	S21 = 1.000 S14 = 1.000	S21 = 1.000 S14 = 1.000	S21 = 1.000 S14 = 1.000	0.782
IdC (%)	−5.78 ± 9.51	4.18 ± 13.46	−6.70 ± 7.05	−2.77 ± 6.87	−2.24 ± 9.58	2.54 ± 7.13	3.80 ± 4.94	1.37 ± 6.38	0.598	0.820	0.03 *	S21 = 0.512 S14 = 1.000	S21 = 1.000 S14 = 0.155	S21 = 0.214 S14 = 1.000	0.236
IVV (%)	23.00 ± 7.57	24.75 ± 3.20	20.75 ± 9.43	22.83 ± 6.01	17.20 ± 6.42	19.00 ± 7.42	17.00 ± 4.00	17.73 ± 5.82	0.253	0.195	0.443	S21 = 1.000 S14 = 1.000	S21 = 0.687 S14 = 1.000	S21 = 0.757 S14 = 1.000	0.799

SR = stroke rate, SL = stroke length, IdC = index of coordination, IVV = intracyclic velocity variation, T = total. Differences between groups are identified by *.

## Data Availability

The original contributions presented in this study are included in the article. Further inquiries can be directed to the corresponding author.

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
