# Peer review of "Some Biomechanical and Anthropmetric Differences Between Elite Swimmers with Down Syndrome and Intellectual Disabilities"

_sports, 2026, doi:10.3390/sports14010028_

Round 1

Reviewer 1 Report

Comments and Suggestions for Authors

Dear,

This manuscript highlights how specific biomechanical, coordinative, and anthropometric differences between swimmers with Down syndrome and those with other intellectual disabilities can directly inform tailored rehabilitation, training, and inclusion strategies. By identifying distinct biomechanical and coordinative limitations, such as disadvantageous anthropometric characteristics (e.g., shorter limbs, smaller hands and feet) and lower swimming velocity and efficiency, the motor challenges specific to these populations can be better understood. These findings support the development of targeted aquatic therapy protocols, adapted physical education strategies, and performance monitoring tools that aim to enhance motor coordination, cardiovascular fitness, and functional independence, ultimately contributing to the improved health and quality of life of individuals with Down syndrome and those with other intellectual disabilities.

This retrospective study is well-designed and methodologically sound, but several aspects require correction or clarification:

The title is inconsistent with the abstract and the main body of the paper. Specifically, the title mentions "Intellectual Disability" in the singular form, whereas the term is used in the plural form ("Intellectual Disabilities") throughout the text. This inconsistency should be corrected and harmonised across all sections.

Furthermore, the aims stated in the abstract do not fully align with those presented at the end of the Introduction chapter. However, it is commendable that the abstract mentions that the tests were conducted at maximum velocity. This important detail must also be emphasised in the main text, as it is currently omitted. Hence, within the M&M chapter, it should be explicitly stated that all tests were performed at maximum movement velocity to maintain consistency with the abstract and to properly reflect the study's design and aims.

In Chapter 2 (M&M), it is essential to emphasise that a 3D kinematic analysis was conducted. A figure describing the 3D position of cameras, pool and subjects is welcome.

Additionally, the authors should cite references supporting the validity and reliability of the software used—Qualisys Track Manager v2.7 and Visual3D (T-Motion, Germantown, MD, USA).

In the Discussion chapter, the limitations of the study should be acknowledged and discussed.

Finally, the manuscript should include a separate and clearly defined Conclusion chapter, which directly addresses the research aims and hypotheses outlined in the Introduction.

Kind regards

Author Response

1. Summary

Thank you very much for taking the time to review this manuscript. Please find the detailed responses below that we hope are in accordance with your expectations.

2. Point-by-point response to Comments and Suggestions for Authors

Comments 1: The title is inconsistent with the abstract and the main body of the paper. Specifically, the title mentions "Intellectual Disability" in the singular form, whereas the term is used in the plural form ("Intellectual Disabilities") throughout the text. This inconsistency should be corrected and harmonised across all sections.

Response 1: Thank you for pointing this out. We agree with this comment and we have changed the title to “Intellectual Disabilities”.

Comments 2: Furthermore, the aims stated in the abstract do not fully align with those presented at the end of the Introduction chapter. However, it is commendable that the abstract mentions that the tests were conducted at maximum velocity. This important detail must also be emphasised in the main text, as it is currently omitted. Hence, within the M&M chapter, it should be explicitly stated that all tests were performed at maximum movement velocity to maintain consistency with the abstract and to properly reflect the study's design and aims.

Response 2: We agree. We have, accordingly, added “characterize and” at the Abstract (line 21), “and coordinative” (line 91) and “at maximum velocity (line 92) in the Introduction. As for the M&M, the text already states that they perfomed at maxmum effort (line113).

Comments 3: In Chapter 2 (M&M), it is essential to emphasise that a 3D kinematic analysis was conducted. A figure describing the 3D position of cameras, pool and subjects is welcome.

Response 3: We agree with the reviewer. We added a figure on the experimental setup in the swimming pool.

Comments 4: Additionally, the authors should cite references supporting the validity and reliability of the software used—Qualisys Track Manager v2.7 and Visual3D (T-Motion, Germantown, MD, USA).

· Response 4: We added “Qualisys is a certified company with ISO 9001 Certificate and EC Certificate – Medical Device Directive (MDD) that can be found at www.qualisys.com.” (Lines 128-130)

Comments 5: In the Discussion chapter, the limitations of the study should be acknowledged and discussed.

Response 5: We added “This is a limitation from the present study that the authors aknowledge. It would be important to analize gender differences. Nevertheless, it is difficult to find a sufficient number of swimmers with Down syndrome and intellectual disabilities that train at a high level to this purpose” (lines 290-293).    

Comments 6: Finally, the manuscript should include a separate and clearly defined Conclusion chapter, which directly addresses the research aims and hypotheses outlined in the Introduction.

Response 6: We agree with your observation and added a new chapter “5. Conclusions”

Reviewer 2 Report

Comments and Suggestions for Authors

sports-3801595

Reviewer comments

In the submitted manuscript, the authors examined the anthropometric and kinematical differences between para-swimmers with Down Syndrome and intellectual impairment. Results indicated that lower swimming efficiency was observed in the para-swimmers with Down syndrome due to disadvantageous anthropometrics, less coordination and increased intracyclic velocity variation.

The submission is well within the scope of the Journal. However, there are several topics that need to be addressed, as mentioned in the following General and Specific Comments.

General Comments

  • Introduction: further elaboration and evidence is required about the effect of the pathology on performance relating Down Syndrome. Also, elaboration is needed on how intellectual impairment affects swimming performance. In addition, further development of the rationale is required for the comparison of the two para-swimming classes. Furthermore, the importance of IVV and IdC is recommended to be further introduced. For example, explanations are presented within the Results section (LL169-172) for these variables.
  • Methods: the sample size is small. Also, there seems to be an imbalance regarding age between the examined groups. In addition, class S21 is not included in the contemporary World Para Swimming Classification Rules and Regulation. Please, clarify.
  • Discussion: It is suggested to update the discussion with concurrent research findings, as only several references were published in the last 5 years. Also, it is suggested to clearly state the limitations of the study.
  • Conclusions: it is recommended to add a section with practical implications for coaches and practitioners focused on the main outcomes that are derived from the present findings.

Specific Comments

Abstract

  • LL24-25: with the Qualysis motion analysis system
  • L26 &L31: it is recommended to replace the semi-colons (;) with full stops (.).

Introduction

  • It is suggested to elaborate on coordination based on i.e.:
    • Nikodelis, T., Kollias, I., & Hatzitaki, V. (2005). Bilateral inter-arm coordination in freestyle swimming: Effect of skill level and swimming speed. Journal of Sports Sciences, 23(7), 737-745.
    • Seifert, L., Komar, J., Barbosa, T., Toussaint, H., Millet, G., & Davids, K. (2014). Coordination pattern variability provides functional adaptations to constraints in swimming performance. Sports medicine, 44(10), 1333-1345.
    • Morais, J. E., Forte, P., Nevill, A. M., Barbosa, T. M., & Marinho, D. A. (2020). Upper-limb kinematics and kinetics imbalances in the determinants of front-crawl swimming at maximal speed in young international level swimmers. Scientific Reports, 10(1), 11683.

Materials and Methods

  • It is suggested to insert subsection headers (i.e., participants, experimental procedure, statistical analysis, etc.).
  • LL98-100: Provide training age, level of competition (i.e., special Olympics or Para-swimming events), and ranking scores.
  • L108: Please, specify where in the test the data acquisition and analysis was conducted (i.e., the last 5 m of the 25 m tests?)
  • LL110-113: Provide details on how the distortion was handled.
  • LL121-122: Provide the description of the anthropometric measures mentioned in the abstract and in Table 1. Also clarify the stadiometer (Seca model 708 is a scale).
  • LL123-124: Provide the instructions given for the period before the body fat measures. Could these measures be conducted with the swimming suit to enhance the ecological validity of this assessment?
  • L128: Clarify which hip was assessed. Since a full body set-up was adopted (LL116-120), what was the rationale to measure swimming velocity by the hip displacement and not based on the center of mass movement?
  • LL148-153: Provide further details, due to the small sample size, if the assumptions for the ANOVA were met. In addition, effect sizes could add context to the presentation of the results and thus it is recommended to include them in the statistical analyses.
  • L151: There is no mention for the kinematic parameters. Please, clarify.
  • L153: Provide the manufacturer details for SPSS.

Results

  • LL165-167: define T in Table 2.
  • LL169-172: see the respective General Comment.

Discussion

  • L178: It is recommended to start the Discussion by explicitly addressing the hypotheses of the study.
  • L203: by the fact

Conclusions

  • See the respective General Comment.

References

  • Provide refs. #23 and #29 according to the Journal style.

Author Response

1. Summary

Thank you very much for taking the time to review this manuscript. Please find the detailed responses below that we hope are in accordance with your expectations.

2. Point-by-point response to Comments and Suggestions for Authors

General Comments 1: Introduction: further elaboration and evidence is required about the effect of the pathology on performance relating Down Syndrome. Also, elaboration is needed on how intellectual impairment affects swimming performance. In addition, further development of the rationale is required for the comparison of the two para-swimming classes. Furthermore, the importance of IVV and IdC is recommended to be further introduced. For example, explanations are presented within the Results section (LL169-172) for these variables.

Response 1: We would like to thank the reviewer for the suggestions and comments. We tried to improve this section and added Lines 54-58 (Intellectual disability is characterized by significant limitations both in intellectual funcioning and adaptative behavior, as expressed in terms of conceptual, social and practical adaptive skills (Kyriakidou et al. 2024). Due to the nature if intellectual disabilities, motor function is also often impaired, requiring tailored physical developmental strategies (Giagazoglou et al. 2012). We also added Lines 72-85 (Studies concerning competitive level persons with intellectual disabilities (also Down syndrome) are still scarce [10,12] and research on the effects of swimming on swimmers with intellectual disabilities is limited. A notable gap in exploring the specific challenges and adaptations required for this population in competitive swimming is observed (Kyriakidou et al. 2024). Although people with intellectual disabilities benefit from improved cardiorespiratory and muscular endurance, speed, balance, and agility in the water, they show distinctive differences in swimming performance and muscular strength (Marques-Aleixo et al. 2013). Persons with Down syndrome are known to have poorer strength, cardiovascular fitness and body composition than persons with intellectual disabilities or non-disabled individuals [13-15] although they are able to improve their physical and functional fitness [11, 16-22]. Also, elite swimmers with intellectual disabilities face challenges in speed, particularly during turns, compared to swimmers without intellectual disabilities (Einarsson et al. 2008). These swimmers often encouter challenges in adopting a strategic approach during competitive events, which can impact adversely on the performance (Kyriakidou et al. 2024).

· General Comments 2: Methods: the sample size is small. Also, there seems to be an imbalance regarding age between the examined groups. In addition, class S21 is not included in the contemporary World Para Swimming Classification Rules and Regulation. Please, clarify.

Response 2: We agree that the sample size is small. It is, in our opinion, one of the greatest limitation of this study. This is mainly due to the difficulty in gathering high level swimmers with Down syndrome and intellectual disabilities. (with sufficient training and competetive experience)

Class S21 is not, in fact, included in the World Para Swiming Classification Rules and Regulation, as they are considered in the Class S14, so they swim at the same Class as swimmers with intellectual disabilities. They might in fact be at a disadvantage in this class.

· General Comments 3: Discussion: It is suggested to update the discussion with concurrent research findings, as only several references were published in the last 5 years. Also, it is suggested to clearly state the limitations of the study.

Response 3: We agree with the reviewer. Nevertheless, studies concerning swimming for down syndrome and intellectual disabilities are still very scarce, especially when it comes to high level. Most of the studies are still with recreational participants. We added lines 290-293 about limitations.

“This is a limitation from the present study that the authors aknowledge. It would be important to analize gender differences. Nevertheless, it is difficult to find a sufficient number of swimmers with Down syndrome and intellectual disabilities that train at a high level to this purpose”.  

Nevertheless, we added some sentences at the discussion, especially related to coordination and IVV from more recent perspectives: “Nevertheless, Silva et al. [34] highlighted the fact that upper-limb coordination varies with organismic, task and environmental constraints and suggest that is no ideal or optimal coordination pattern that youth, learners, and less skilled swimmers should imitate. They suggest that several motor solutions should be adopted based on how each swimmer deals with specific constraints. Obviously, this raises the question weather swimmers with intellectual disabilities are able to identify and recognize these constraints.” (Lines 305-311).

IVV and IdC provide relevant complementary information to differenciate swimmers according to their skill level [37,40]. (Lines 315-317).

· General Comments 4: Conclusions: it is recommended to add a section with practical implications for coaches and practitioners focused on the main outcomes that are derived from the present findings.

· Response 4: We added a new Chapter “5. Conclusions” with a new paragraph.

“It is important that coaches aknowledge the importance of these biomechanical and coordinative parameters and use them when preparing their training sessions. Swimmers with Down syndrome and intellectual disabilities should have their SR and SL relationship optimized by training and it is important, especially for Down syndrome swimmers that, coordination and IVV should not be neglected by coaches, as they may make a difference regarding swimming velocity.” (lines 311-316.

Specific Comments 1: Abstract: LL24-25: with the Qualysis motion analysis system. L26 &L31: it is recommended to replace the semi-colons (;) with full stops (.).

Response 1: Changed in the manuscript.

Specific Comments 2: Introduction It is suggested to elaborate on coordination based on i.e.: Nikodelis, T., Kollias, I., & Hatzitaki, V. (2005). Bilateral inter-arm coordination in freestyle swimming: Effect of skill level and swimming speed. Journal of Sports Sciences, 23(7), 737-745. Seifert, L., Komar, J., Barbosa, T., Toussaint, H., Millet, G., & Davids, K. (2014). Coordination pattern variability provides functional adaptations to constraints in swimming performance. Sports medicine, 44(10), 1333-1345. Morais, J. E., Forte, P., Nevill, A. M., Barbosa, T. M., & Marinho, D. A. (2020). Upper-limb kinematics and kinetics imbalances in the determinants of front-crawl swimming at maximal speed in young international level swimmers. Scientific Reports, 10(1), 11683.

Response 2: We would like to thank the reviewer for the precious suggestions. We added several sentences at Introduction concerning coordination:

“Although coordination has not been directly related to swimming propulsion, its connection to buoyancy, breathing and minimizing drag is fundamental to optimising propulsive forces [32]. Studies have shown that when increasing their speed, swimmers should favour continuity between the left and right upper-limbpropulsive phases, so they couls promote temporal continuity in propulsions (e.g. [29, 33]). Swimmers should also be able to adapt their coordination to minimise energy cost [34]”. (Lines91-97)

“... since higher temporal continuity between propulsive action minimises the decelaration between two propusive cycles [36]. Also, maintaining a lower IVV helps to decrease the energy expenditure [27]”. (Lines 101-104)

Specific Comments 3: Materials and Methods: It is suggested to insert subsection headers (i.e., participants, experimental procedure, statistical analysis, etc.).

· LL98-100: Provide training age, level of competition (i.e., special Olympics or Para-swimming events), and ranking scores.

· L108: Please, specify where in the test the data acquisition and analysis was conducted (i.e., the last 5 m of the 25 m tests?)

· LL110-113: Provide details on how the distortion was handled.

· LL121-122: Provide the description of the anthropometric measures mentioned in the abstract and in Table 1. Also clarify the stadiometer (Seca model 708 is a scale).

· LL123-124: Provide the instructions given for the period before the body fat measures. Could these measures be conducted with the swimming suit to enhance the ecological validity of this assessment?

· L128: Clarify which hip was assessed. Since a full body set-up was adopted (LL116-120), what was the rationale to measure swimming velocity by the hip displacement and not based on the center of mass movement?

· LL148-153: Provide further details, due to the small sample size, if the assumptions for the ANOVA were met. In addition, effect sizes could add context to the presentation of the results and thus it is recommended to include them in the statistical analyses.

· L151: There is no mention for the kinematic parameters. Please, clarify.

· L153: Provide the manufacturer details for SPSS.

· Response 3: Subsection headers added. Added some information on level of competition. The added figure must respond to the doubt. The measures wer made with swimmers wearing light clothes and no shoes. Both right and left hips were assessed.  

Specific Comments 4: Results

· LL165-167: define T in Table 2.

· LL169-172: see the respective General Comment.

Response 4: The T was defined.

Specific Comments 5: Discussion: L178: It is recommended to start the Discussion by explicitly addressing the hypotheses of the study. L203: by the fact

Response 5: The hypotheses of the study were adressed at the first paragraph of the Discussion, as suggested.

“The main hypothesis of the present study were: (i) swimmers with intellectual disabilities are faster than those with Down syndrome; (ii) swimmers with Down syndrome present higher SR and SL values than intellectual disabled peers; (iii) swimmers with Down syndrome apply different coordination modes and present higher IVV and intra-variability than swimmers with intellectual disability for velocity, SR, SL, IdC and IVV.” (Lines 186-191).

Specific Comments 6: Conclusions: See the respective General Comment.

Response 6: Response above.

Specific Comments 7: References: Provide refs. #23 and #29 according to the Journal style.

Response 7: Provided on the manuscript.

Reviewer 3 Report

Comments and Suggestions for Authors

The article presents a description of the kinematic differences between swimmers with Down syndrome and swimmers with intellectual disabilities. While the topic is interesting, I believe the article misses its target, or rather, that the title is not appropriate. Indeed, whereas one would expect to find detailed information on the athletes’ kinematics, more than half of the article is devoted to purely anthropometric differences. The rest merely touches on a few parameters—certainly related to the athletes' kinematics—but mostly used as markers of general performance. Overall, I think the focus of the article should be readjusted, and the choice of terminology should be more consistent throughout: kinematics, biomechanics, performance markers, swimming performance, technical characteristics, etc.

Here are a few more specific points:

  • Nearly half of the abstract is dedicated to exhaustive details about anthropometric differences. This is quite confusing given that one expects a kinematic analysis. This should be revised after refocusing the article’s objective.

  • The introduction also needs improvement. The literature review is quite weak, even though the authors mention that little literature exists on the mentioned conditions. However, there are many other articles on swimming that use the same techniques. A brief literature overview would have been appropriate. More generally, the research question is not clearly articulated. What studies have been conducted in swimming, involving people with Down syndrome, intellectual disabilities, or other disabilities? The review is very focused on Down syndrome but says very little about intellectual disabilities and doesn't clearly show what is lacking in the literature. Why is this study necessary?

  • Furthermore, the stated aim does not really match the title, since the focus is on biomechanical determinants of performance using a kinematic approach. The second objective is unclear or too vague: how will you contribute to the development of competitive athletes?

  • The methodology is relatively well written. However, the warm-up protocol and the experimental setup should be described in more detail. An illustration of the experimental setup would be helpful, as well as a justification for choosing the 7.5m and 20m markers. Also, why prevent the swimmers from breathing when they normally do? Doesn’t that distort their swimming style? A reference or explanation for the type of filter used would be necessary. Finally, it is a pity to have such a complete and complex setup, with multiple markers allowing for 3D kinematic analysis, only to barely address it and discuss only a few performance parameters that could technically be measured using simple video footage.

  • In the Results section, be careful not to include interpretations, such as "clear disadvantage," for example. In the tables, significant differences could be highlighted in bold and marked with an asterisk (*) instead of an "a."

  • Lastly, in the Discussion section, greater caution should be exercised when making assertions. For instance, you cannot claim that swimmers with Down syndrome have a clear disadvantage compared to others, given the very small sample size. This is, in fact, the second major limitation of your study. Nine athletes—four and five per group—is extremely small and severely limits the study’s generalizability. Furthermore, again, a lot of space is given to anthropometric differences rather than kinematic data, as suggested by the title. Lines 227–237: I don’t really see the link with the study, other than the shared disability being discussed? And I don’t understand the phrase "clear advantage of the present study."

  • No conclusion?

Author Response

1. Summary

Thank you very much for taking the time to review this manuscript. Please find the detailed responses below that we hope are in accordance with your expectations.

2. Point-by-point response to Comments and Suggestions for Authors

· General Comments: The article presents a description of the kinematic differences between swimmers with Down syndrome and swimmers with intellectual disabilities. While the topic is interesting, I believe the article misses its target, or rather, that the title is not appropriate. Indeed, whereas one would expect to find detailed information on the athletes’ kinematics, more than half of the article is devoted to purely anthropometric differences. The rest merely touches on a few parameters—certainly related to the athletes' kinematics—but mostly used as markers of general performance. Overall, I think the focus of the article should be readjusted, and the choice of terminology should be more consistent throughout: kinematics, biomechanics, performance markers, swimming performance, technical characteristics, etc.

Response 1: Thank you for your general comments. We understand your concerns. We changed the Title to “Some biomechanical and anthropmetric differences between elite swimmers with Down syndrome and intellectual disabilities“. Concerning the anthropometric analysis there are two points we would like to enhance: 1) the importance of the anthropometric parameters for swimming, namely the arm span, height, hand and feet dimensions. As persons with Down syndrome usually have distinctive anthropometric characteristics, that could impact on swimming performance, it was our intention to analyze those results and compare them to swimmers with intellectual disabilities, that usually have no other impact than the lower IQ. 2) The lack of studies concerning high trained swimmers with Down syndrome and intellectual disabilities that highlight the importance of describing and analizing some data that was already studied for athletes without disabilities years ago.

· Specific Comments 1: Nearly half of the abstract is dedicated to exhaustive details about anthropometric differences. This is quite confusing given that one expects a kinematic analysis. This should be revised after refocusing the article’s objective.

· Response 1: We agree with the reviewer and made changes accordingly.

· Specific Comments 2: The introduction also needs improvement. The literature review is quite weak, even though the authors mention that little literature exists on the mentioned conditions. However, there are many other articles on swimming that use the same techniques. A brief literature overview would have been appropriate. More generally, the research question is not clearly articulated. What studies have been conducted in swimming, involving people with Down syndrome, intellectual disabilities, or other disabilities? The review is very focused on Down syndrome but says very little about intellectual disabilities and doesn't clearly show what is lacking in the literature. Why is this study necessary? 

Response 2: We would like to thank the reviewer for the suggestions and comments. We tried to improve this section and added Lines 53-56 (“Intellectual disability is characterized by significant limitations both in intellectual funcioning and adaptative behavior, as expressed in terms of conceptual, social and practical adaptive skills [8]. Due to the nature if intellectual disabilities, motor function is also often impaired, requiring tailored physical developmental strategies [9].”)

We also added Lines 70-82 (“... and research on the effects of swimming on swimmers with intellectual disabilities is limited. A notable gap in exploring the specific challenges and adaptations required for this population in competitive swimming is observed [8]. Although people with intellectual disabilities benefit from improved cardiorespiratory and muscular endurance, speed, balance, and agility in the water, they show distinctive differences in swimming performance and muscular strength [15]. Persons with Down syndrome are known to have poorer strength, cardiovascular fitness and body composition than persons with intellectual disabilities or non-disabled individuals [16-18] although they are able to improve their physical and functional fitness [13, 19-25]. Also, elite swimmers with intellectual disabilities face challenges in speed, particularly during turns, compared to swimmers without intellectual disabilities [26]. These swimmers often encouter challenges in adopting a strategic approach during competitive events, which can impact adversely on the performance [8].)   

· Specific Comments 3: Furthermore, the stated aim does not really match the title, since the focus is on biomechanical determinants of performance using a kinematic approach. The second objective is unclear or too vague: how will you contribute to the development of competitive athletes?

· Response 3: In order to make the aim clearer, we changed the title. With the second objetive it is our intention to contribute to the characterization of high level swimmers with intellectual disabilities and Down syndrome, an area of study that is clearly in need of development. We are also interested in better understandig differences between these two groups of swimmers, as they compete together at S14 Class at Paralympic competitions.

· Specific Comments 4: The methodology is relatively well written. However, the warm-up protocol and the experimental setup should be described in more detail. An illustration of the experimental setup would be helpful, as well as a justification for choosing the 7.5m and 20m markers. Also, why prevent the swimmers from breathing when they normally do? Doesn’t that distort their swimming style? A reference or explanation for the type of filter used would be necessary. Finally, it is a pity to have such a complete and complex setup, with multiple markers allowing for 3D kinematic analysis, only to barely address it and discuss only a few performance parameters that could technically be measured using simple video footage.

Response 4: We would like to thank the reviewer for the suggestions. We added a brief description of the warm up (Line 112) and a Figure of the experimental setup (Line 122). Concerning breathing, several studies on swimmers without disabilities were made without brething. It is a very good suggestion for future studies. Also, it is usual for swimmers not to breath at smaller distances to optimize swimming velocity.

· Specific Comments 5: In the Results section, be careful not to include interpretations, such as "clear disadvantage," for example. In the tables, significant differences could be highlighted in bold and marked with an asterisk (*) instead of an "a."

Response 5: Thank you for your comments. Changed as suggested.

· Specific Comments 6: Lastly, in the Discussion section, greater caution should be exercised when making assertions. For instance, you cannot claim that swimmers with Down syndrome have a clear disadvantage compared to others, given the very small sample size. This is, in fact, the second major limitation of your study. Nine athletes—four and five per group—is extremely small and severely limits the study’s generalizability. Furthermore, again, a lot of space is given to anthropometric differences rather than kinematic data, as suggested by the title. Lines 227–237: I don’t really see the link with the study, other than the shared disability being discussed? And I don’t understand the phrase "clear advantage of the present study."

Response 6: We would like to thank the revirewer for the comments. We changed the refered assertions. We alsdo added a few statements mostly on coordination:

“Nevertheless, Silva et al. [34] highlighted the fact that upper-limb coordination varies with organismic, task and environmental constraints and suggest that is no ideal or optimal coordination pattern that youth, learners, and less skilled swimmers should imitate. They suggest that several motor solutions should be adopted based on how each swimmer deals with specific constraints. Obviously, this raises the question weather swimmers with intellectual disabilities are able to identify and recognize these constraints.” (Lines 305-311).

IVV and IdC provide relevant complementary information to differenciate swimmers according to their skill level [37,40]. (Lines 315-317).

Specific Comments 7: No conclusion?

Response 7: We added a new Chapter “5. Conclusions”.

Round 2

Reviewer 2 Report

Comments and Suggestions for Authors

sports-3801595-R1

Reviewer comments

In the resubmitted manuscript, the authors did an exceptional work to address the concerns raised in the initial round of reviewing. Nevertheless, there are still some topics that need to be addressed, as mentioned in the following Comments.

Comments

  • Class S21 is still mentioned.
  • Further details on how the distortion was handled (the photograph actually enhances the concern about this methodological issue).
  • Information is still missing regarding the point of the test where calibration and measurement was conducted.
  • Seca model 708 is a scale; if a stadiometer was fixed to it, please describe the model of the stadiometer.
  • Please, clarify how both right and left hip data were managed to provide swimming velocity.
  • Statistical analysis: Provide further details, due to the small sample size, if the assumptions for running the ANOVA were met, set the effect size thresholds and include the manufacturer details for SPSS.

Author Response

1. Summary

Thank you very much for taking the time to review this manuscript and provide your comments and suggestions for improvements. Please find the detailed responses below that we hope are in accordance with your expectations.

2. Point-by-point response to Comments and Suggestions for Authors

· Comments 1: Class S21 is still mentioned.

Response 1: We acknowledge the fact that Class S21 is still mentioned, mainly due to: 1) it is the most direct way of dintinguish these swimmers from swimmers with intellectual disabilities (S14), since it is our intention to compare both groups of swimmers; 2) as it is reffered in the manuscript, S21 is not a paralympic Class, but in fact some Countries use this designation for swimmers with Down syndrome and there are International Events promoted by Down Syndrome International Swimming Organization (DSISO) that use this designation.

· Comments 2: Further details on how the distortion was handled (the photograph actually enhances the concern about this methodological issue).

Response 2: Prior to testing, calibration process was conducted to minimize also distortion. We rephrased the paragraph in order to try to make it more informative.

“The calibrated volume was defined using three calibrations – underwater, overwater and twin (to merge the first and the latter) – according to manufacturer’s guidelines. This approach enabled the creation of 3D dual-media volume, where the orthogonal axes were defined as x, y, and z for horizontal, mediolateral, and vertical movements (z = 0 defines water surface). A 13-camera setup (Motion Capture, Qualysis Cameras; Qualisys AB, Gothenburg, Sweden) was used, with seven dry-land plus six underwater cameras (Oqus 3+ and Oqus Underwater; Qualisys AB, Gothenburg, Sweden) operating at 100 Hz” (lines 137-144).

· Comments 3: Information is still missing regarding the point of the test where calibration and measurement was conducted.

Response 3: Calibration was conducted prior to any test. It was conducted according to manufacturer’s guidelines, as reffered in the text.

· Comments 4: Seca model 708 is a scale; if a stadiometer was fixed to it, please describe the model of the stadiometer.

Response 4: The model of the stadiometer was Seca 213 (Seca GmbH & Co. KG, Hamburg, Germany), which was added to the manuscript.

· Comments 5: Please, clarify how both right and left hip data were managed to provide swimming velocity.

Response 5: We added an information that was missing in the manuscript. Swimming velocity was calculated from the displacement of the right hip (line 163).

· Comments 6: Statistical analysis: Provide further details, due to the small sample size, if the assumptions for running the ANOVA were met, set the effect size thresholds and include the manufacturer details for SPSS.

Response 6: The assumptions for running the ANOVA were met. Normality was checked by the Shapiro-Wilk test and homogeneity of variance by Levene’s test. Also, despite being a small sample size, it was specific. More details were added to the manuscript.
